# Metasurface Spiral Focusing Generators with Tunable Orbital Angular Momentum Based on Slab Silicon Nitride Waveguide and Vanadium Dioxide (VO_2_)

**DOI:** 10.3390/nano10091864

**Published:** 2020-09-17

**Authors:** Li Chen, Lin Zhao, Yuan Hao, Wenyi Liu, Yi Wu, Zhongchao Wei, Ning Xu, Shuai Qin, Xiangbo Yang, Hongzhan Liu

**Affiliations:** Guangdong Provincial Key Laboratory of Nanophotonic Functional Materials and Devices, School for Information and Optoelectronic Science and Engineering, South China Normal University, 378 Waihuan West Road Panyu District, Guangzhou 510006, China; 2019022102@m.scnu.edu.cn (L.C.); 2017021792@m.scnu.edu.cn (L.Z.); 2018022081@m.scnu.edu.cn (Y.H.); 2019022111@m.scnu.edu.cn (W.L.); wy1121377742@gmail.com (Y.W.); wzc@scnu.edu.cn (Z.W.); xuning199405@outlook.com (N.X.); 2019022112@m.scnu.edu.cn (S.Q.); xbyang@scnu.edu.cn (X.Y.)

**Keywords:** vanadium dioxide, phase change material, orbital angular momentum, metasurface

## Abstract

The metasurface spiral focusing (MSF) generator has gained attention in high-speed optical communications due to its spatial orthogonality. However, previous MSF generators only can generate a single orbital angular momentum (OAM) mode for one polarized light. Here, a MSF generator with tunable OAM is proposed and it has the ability to transform linearly polarized light (LPL), circularly polarized light or Gaussian beams into vortex beams which can carry tunable OAM at near-infrared wavelength by controlling the phase transition of vanadium dioxide (VO_2_). Utilizing this MSF generator, the beams can be focused on several wavelength-sized rings with efficiency as high as 76%, 32% when VO_2_ are in the insulating phase and in the metallic phase, respectively. Moreover, we reveal the relationship between the reflective focal length and transmissive focal length, and the latter is 2.3 times of the former. We further demonstrate the impact of Gaussian beams with different waist sizes on MSF generators: the increase in waist size produces the enhancement in spiral focusing efficiency and the decrease in size of focal ring. The MSF generator we proposed will be applicable to a variety of integrated compact optical systems, such as optical communication systems and optical trapping systems.

## 1. Introduction

It is well known that electromagnetic waves can carry angular momentum. Angular momentum comprises spin angular momentum (SAM) and orbital angular momentum (OAM) [1,2]. Beth [1] initially observed that circularly polarized light has SAM of ±h per photon (h is reduced Planck’s constant) and SAM is associated with the polarization of electromagnetic waves. Dissimilar to the SAM, Allen [2] recognized that OAM is related to the spatial phase distribution of electromagnetic waves, and vortex beams carrying OAM are characterized by a rotating phase factor of exp(ilθ), where θ refers to the azimuthal angle and *l* is topological charge. It’s worth noticing that *l* is an unbounded integer and vortex beams with different OAM modes are mutually orthogonal [1]. Therefore, without expanding the frequency band, OAM modes are used for encoding information in order to realize multiplexing and improve the channel capacity of communication system [3,4,5,6,7,8]. Appearing in the last decades, vortex beams with OAM have been extended to countless applications, such as biosciences [9], data storage [10,11], quantum information processing [12] and so on.

The devices for generating vortex beams have been extensively investigated, including spiral phase plates [13,14,15], antenna arrays [16,17,18], spatial light modulators [19,20], geometric mode converters [21], etc. These traditional devices for generating vortex beams are always too bulky to be integrated into nanoscale optoelectronic systems. In the past ten years or more, metamaterials have been increasingly developed in manipulating electromagnetic waves, and have provided new approaches to achieve characters not existing in natural materials [22,23,24,25]. Recently, 2D metamaterials (also called metasurfaces), two-dimensional artificial structures consisting of arrays of subwavelength metal or dielectric optical elements with periodic arrangement, have obtained an increasing popularity for their special capability of flexibly manipulating the amplitude, polarization and phase of the transmitted or reflected electromagnetic waves at nanoscale [26,27,28,29,30,31,32,33,34,35]. So far, there have been significant works in the development of metasurfaces for generating vortex beams [36,37,38,39,40,41,42,43,44]. However, these works usually fail to achieve adjustable topological charge, which means that, once it is manufactured, its topological charge is fixed. Besides, their effective operation range is mainly limited to the visible light or mid-infrared, and their light sources rely deeply on circularly polarized light or linearly polarized light (LPL). Therefore, these limitations will increase the complexity of the experimental measurement device or photonic integrated system. At the same time, the impact of Gaussian beams with different waist sizes on the MSF generator to produce spiral focusing beams (SFB) has not been demonstrated before.

Based on the phase principle of the slab waveguide theory and the phase change material vanadium dioxide (VO_2_), we numerically design a highly efficient, polarization-insensitive MSF generator to produce SFB with adjustable OAM, which can generate SFB with topological charge *l* in the telecom waveband (1550 nm) as VO_2_ is in the insulating phase. When the actual temperature is above the phase transition temperature (340 K), VO_2_ becomes metallic-phase, and the transmissive metasurface can turn into reflective metasurface with strong absorption, and the SFB carrying topological charge 2*l* are produced in the telecom waveband (1550 nm). Our device can overcome the shortcomings of previous devices that generate optical vortices to a certain extent. The MSF generator firstly produces the exiting light with a helical phase wavefront on the reflection surface or transmission surface, and then the vortex beams be focused on several wavelength-sized rings whose efficiency is up to 76% and 32% for transmission mode and reflection mode, and the transmissive focal length is 2.3 times of the reflective focal length. In addition, we further demonstrate the impact of Gaussian beams with different waist sizes on the MSF generator: the increase in waist size produces the enhancement in spiral focusing efficiency and the decrease in size of focal ring. The MSF generator with tunable OAM has potential application in optical manipulation at nanoscale [45] and optical fiber communication of optical metasurfaces [46].

## 2. Theory and Design

Figure 1a,b shows the schematic diagram of SFB carrying topological charge *l* = 2 in transmission mode and *l* = 4 in reflection mode, respectively. In transmission mode, its phase wavefront (Figure 1c) may be considered as the result of superimposition of the spiral phase distribution $\varphi_{1}=l\theta$ (Figure 1d) along the azimuthal direction that produces OAM, and the parabola phase distribution $\varphi_{2}=k\sqrt{f^{2}+r^{2}}$ (Figure 1e) along the radial direction that focuses vortex beams on the desired focal length. In reflection mode, its phase wavefront can be regarded as the result of superimposition of $2\varphi_{1}$ and $2\varphi_{2}$. Figure 1f is a schematic diagram of the metasurface with tunable OAM. It is made up of thousands of square silicon nitride (SSN) nanopillars, square silicon dioxide with thickness of subwavelength scale and VO_2_. The phase of SSN nanopillars in the transverse plane $\left(x_{i},y_{i}\right)$ are expressed by:(1)$$\varphi (x_{i},y_{i},\theta )=2m\pi +\frac{2\pi }{\lambda_{i}}(ƒ-\sqrt{{ƒ}^{2}+x_{i}^{2}+y_{i}^{2}})+l\theta$$
where *m* represents an arbitrary integer number, $\lambda_{i}$ denotes the design wavelength and *f* is the focus length of the MSF generator, *l* represents topological charge and θ refers to the azimuthal angle at transverse plane $\left(x_{i},y_{i}\right)$. The details of the unit cell structure are plotted in Figure 1g,h, where a SSN nanopillar with height (*H*) placed on substrate composed of VO_2_ and silicon dioxide.

VO_2_ is a typical metal–insulator transition material. The temperature hardly affect the optical permittivity of VO_2_ when the temperature is far away from the phase transition point (340 K) [47]. VO_2_ insulator material can be regarded as dielectric. Within 1 um to 5 um, the dielectric constant of VO_2_ insulator material is approximately 9 [47], as presented in Figure 2a. VO_2_ metal material corresponds to plasmonic and is expressed by the Drude model with the permittivity at high frequency ($\varepsilon_{\infty }\;=\;3.95$), plasma frequency ($\omega_{\rho }\;=3.33\;\mathrm{ev}$) and scattering rate (γ = 0.66 ev) [48]. To obtain the dielectric constant of the VO_2_ metal material, we substituted the experimental data in [48] into the classic Drude-Lorentz oscillator model, as shown in Figure 2b. The advantage of using silicon nitride as the nanopillar and silicon dioxide as the substrate is that their thermo-optic coefficient is very small [49,50], which means that the switch of VO_2_ between insulator phase and metallic phase does not significantly change the refractive index of SSN nanopillars. 

To further discover the phase realization mechanism, the numerical simulation of the unit cell structure is performed using the commercially available three-dimensional finite difference time domain (FDTD) solver from Lumerical Inc (Lumerical Inc., Vancouver, BC, Canada). The size of a mesh cell is 0.01 µm. The relationship between the effective index of single SSN nanopillar and its width for x-LPL incidence are plotted in Figure 3a. The result denotes that by adjusting the width of SSN nanopillar, the effective index of its fundamental mode can vary anywhere from $n_{eff}\approx n_{air}=1$ (when the incident light is mostly in air) to $n_{eff}\approx n_{Si3N4}=2.46$ (when the incident light is mostly in SSN). The effective index of the fundamental mode is changed to gain the desired phase distribution (Equation (1)). If there are two parallel SSN nanopillars, they have different widths and the optical coupling between them is negligible, then light traveling along the different SSN nanopillars will accumulate a phase shift $\Delta \varphi_{WG}$, which is proportional to their height (*H*) [51]: (2)$$\Delta \varphi_{WG}=\frac{2\pi }{\lambda_{i}}\Delta n_{eff}H$$
where $\Delta n_{eff}$ is the difference of effective index between the two SSN nanopillars. A phase difference of $\Delta \varphi_{WG}=2\pi$ can be acquired when $H=\lambda_{i}/\Delta n_{eff}$. Figure 3b displays the appearance of the phase difference among the waveguides of two SSN nanopillars with different widths. It is noteworthy that the waveguide of SSN nanopillar can confine incident light in a subwavelength area owing to large index difference between SSN and air.

To ensure high transmittivity and cover the entire phase span of 0 to 2π, other parameters such as unit cell size (*P*) are optimized in the telecom waveband (1550 nm), as shown in Figure 4a. The height of the SSN nanopillar should be tall and sufficient for covering the entire phase span of 0 to 2π through a range of square widths. Figure 4b denotes that when VO_2_ is in the insulating state and its unit cell size is 652.5 nm, for SSN nanopillars with height of 1.65 µm, their width must be in the range of 250 nm to 566 nm in order to achieve large transmittivity and cover the phase span of 0 to 2π. Figure 4c reveals the reflectivity and phase of the reflected light when VO_2_ is in its metallic state. It can be found that the phase has doubled when VO_2_ changes from insulating state to metallic state by comparing Figure 4b,d. This phenomenon is mainly due to the propagation of reflected light in the SSN waveguide. The discontinuities that appear on the spectra of Figure 4a,c is due to the optical resonance of SSN nanopillar. In the unit cell structure for simulation, periodic boundary conditions are used at the x-boundaries and y-boundaries, and the boundary conditions of perfectly matched layer (PML) are applied at the z-boundaries.

## 3. Results and Discussion

### 3.1. Generation of Spiral Focusing with Adjustable Orbital Angular Momentum

In this section, the MSF generator carrying topological charges of *l* = 2 (in transmission mode) and *l* = 4 (in reflection mode) is demonstrated. The schematic of the MSF generator with the diameters of 40 µm and the focal length (*f*) of 41 µm is shown in Figure 1f. Here, the incident light is x-LPL ($\lambda_{i}=1550\;\mathrm{nm}$). Figure 5a,b are the simulated intensity profiles of y–z cross section in reflection mode and in transmission mode, respectively, and the dashed lines (Figure 5b) represents the position of focal plane (z = −39 µm). The slight difference between the designed focal length and the simulated focal length is caused by the discontinuous phase distribution of the SSN arrays. Figure 5c,d represents the simulated phase distribution of reflected light and transmitted light at one wavelength away from exit facet of the MSF generator, respectively. We can see that the phase distribution in transmission mode corresponded to the calculated phase distribution $\varphi_{1}+\varphi_{2}$ (Figure 1c), and the phase distribution in reflection mode corresponded to the calculated phase distribution $2\varphi_{1}+2\varphi_{2}$ (the inset in Figure 1f). Figure 5e,f depicts the simulated intensity profiles of reflected light and transmitted light on focal plane, respectively. The inset in Figure 5e,f depict the phase distribution of reflected light and transmitted light on focal plane, respectively. We found that SFB carrying topological charge *l* = 2, *l* = 4 may be produced in transmission mode and in reflection mode, respectively. The results of numerical simulation show that spiral focusing efficiency reaches 76% (*l* = 2) and 32% (*l* = 4), which is calculated as the ratio of the optical power in the focal spot region ($P_{ring}$) to that of the incident beam ($P_{incidence}$):(3)$$\eta =\frac{P_{ring}}{P_{incidence}}=\frac{\int_{s_{1}}real(P_{1})dS}{\int_{s_{2}}real(P_{2})dS}$$
where $P_{1}$ is the Poynting vector on focal plane, $S_{1}$ is the area with the same size as the focal ring, $P_{2}$ is the Poynting vector for 1.55 um away from exit facet of the metasurface. $S_{2}$ is the area with the same size as the designed. Figure 5g,h shows the vertical cut of focal ring in reflection mode and in transmission mode, respectively. The full width at half maximum (FWHM) of vertical cut of focal ring are 1.07 µm and 1.58 µm in the two modes.

### 3.2. The Relationship between the Focal Lengths in Two Modes

Here, we discuss the relationship between the focal lengths of the MSF generator in two modes: transmission mode (*l* = 2) and reflection mode (*l* = 4). Figure 6a,b represents the simulation results of intensity profiles on the y-z cross section for MSF generators with different focal lengths in two modes, respectively. The simulation results of FWHM and spiral focusing efficiency for several MSF generators with *f* ranging from 32 to 44 µm in two modes are presented in Figure 6c. It is obvious that the MSF generator with *f* = 41 µm has the spiral focusing efficiency of 76% and 32% in transmission mode and in reflection mode, respectively. It is notable that the focus length in two modes can approximately satisfy the following relationship:(4)$${ƒ}_{1}\approx 2.3{ƒ}_{2}$$
where ${ƒ}_{1}$ is focus length in transmission mode and ${ƒ}_{2}$ is focus length in reflection mode. Obviously, for *l* = 0, the SFB turns into focusing beams with parabola phase distribution ($\varphi_{2}$). Here we design a metasurface focusing (MF) generator with the diameters of 40 µm and focal length (*f*) of 65 µm to generate focusing beams. As found from Figure 6d, the focusing efficiency reaches 72% and 42% in transmission mode and in reflection mode, respectively. We found that the relationship between focus lengths in two modes can approximately satisfy Equation (4), which is because the phase of reflected light can be regarded as $2\varphi_{2}$. Figure 6e shows the simulated values of FWHM and focusing efficiency for several MF generators with *f* ranging from 25 to 45 µm in two modes. It is clear that the MF generator with *f* = 41 µm has a maximum focusing efficiency of 77% in transmission mode, and within a certain range, as the design focal length increases, the focusing efficiency in reflection mode will be enhanced, which is because the simulated phase of the reflected light is closer to $2\varphi_{2}$.

### 3.3. Impact of Different Light Sources on MSF Generator

It is very important to study the influence of different light sources on metasurfaces. If a metasurface can work normally under any light source, it will inevitably reduce the complexity of the experimental measurement device. To understand the impact of different light sources on the MSF generator, we simulate the MSF generator under illumination by plane wave carrying different polarization in transmission mode, the results are shown in Figure 7a–c. For incident plane waves carrying different polarization, the simulated intensity profiles change little, and the corresponding simulation spiral focusing efficiencies are all 76%. That is to say, the spiral focusing effect of our MSF generator is insensitive to incident polarization. The insets in Figure 7a–c display the phase distribution at the focal plane. Besides, the impact of Gaussian beam with different waist sizes on the MSF generator working in transmission mode is demonstrated in Figure 7d. The attenuation of the intensity on the y–z cross section is mainly because the incident power is proportional to the waist size of the Gaussian beams. We can find that the SFB evolves into vortex beams when the waist size of the Gaussian beams decreases. The insets in Figure 7d depict the phase distribution at the focal plane. Figure 7e shows the simulated values of FWHM and spiral focusing efficiency for the incidences of the Gaussian beams of different waist sizes ($W_{0}$ ranging from 12 to 20 µm). It is notable that an increase in waist size produces an enhancement in spiral focusing efficiency and a decrease in the size of FWHM.

### 3.4. Generation of Vortex Beam with Adjustable Orbital Angular Momentum

To further demonstrate the ability of our metasurface to flexibly manipulating phase of the transmitted or reflected light at nanoscale, we design a vortex beams generator with the radii of 30 µm, which has the capability to convert incident Gaussian beams into vortex beams carrying topological charge *l* = 2, *l* = 4 in transmission mode and reflection mode, respectively. As presented in Figure 8. The phase distribution on metasurface for generating vortex beams with *l* = 2 can be designed as $\varphi_{1}=2\theta$ (Figure 8a). Figure 8b shows the simulated transmitted intensity profiles for the vortex beams carrying *l* = 2 on the x–z cross section. The transmitted intensity and phase profiles of the generated vortex beam at 18.6 µm (12 wavelengths), 62 µm (40 wavelengths) and 124 µm (80 wavelengths) away from the exit facet of the metasurface are provided in Figure 8c,d, respectively. We find that the generated vortex beams can transmit a distance of 250 µm and have a divergence angle of 4°. Generally speaking, the propagation distance of vortex beams is proportional to the energy of the incident beam, and the degree of curvature of the equal phase line is proportional to the propagation distance. Our simulation results basically conform to these laws. The metasurface we proposed will operate in reflection mode when VO_2_ changes from insulating state to metallic state, which can convert Gaussian beams into vortex beams carrying topological charge *l* = 4. The topological charge is doubled mainly because the reflected wavefront satisfies $2\varphi_{1}$ (Figure 8e). Figure 8f shows the simulated reflected intensity profiles of the vortex beams carrying topological charge *l* = 4 on the x–z cross section. The simulated reflected intensity and phase distribution of generated vortex beams at 18.6 µm (12 wavelengths), 62 µm (40 wavelengths) and 124 µm (80 wavelengths) away from the exit facet of the metasurface are shown in Figure 8g,h. We notice that the generated vortex beams can only transmit a distance of 150 µm and have a divergence angle of 5.7°. These phenomena are mainly because that the propagation of incident light cannot be completely restricted in the waveguide and the reflection of incident light is incomplete. Moreover, the nonuniform reflectivity of unit cells will contribute to the nonuniform intensity distributions along the angle direction.

## 4. Experimental Feasibility

The experimental measurement is shown in Figure 9. The phase state of VO_2_ can switch between insulating and metallic phase by controlling temperature. The intensity of the MSF generator in transmission mode and reflection mode can be measured through a detector. This experimental measurement can be divided into two parts, the first case, we can make VO_2_ be in the insulating state by controlling room temperature. The incident beam was converted into linearly polarized light with wavelength $\lambda_{i}=1550\;\mathrm{nm}$ through a polarizer. The linearly polarized light was collimated by a collimator, and the beam was split by beam splitter, and then illuminated onto the sample. The generated transmission light was converted into linearly polarization light with wavelength $\lambda_{i}=1550\;\mathrm{nm}$ after passing through a collimator and polarizer. The intensity profiles of the transmitted light were captured by a detector. In the second case, when the MSF generator was located at a temperature controller, we could make VO_2_ in the metallic state by controlling the temperature with the temperature controller. The first half of the measurement process for the second case was the same as the first case. The difference was that the sample at this time was in reflection mode. The intensity profiles of the reflected light can be measured by another detector after passing through the beam splitter, mirror, collimator and polarizer.

## 5. Conclusions

In this article, a highly efficient MSF generator with insensitivity to polarization is proposed, which can realize metallic switches and dielectric metasurfaces. The MSF generator is able to convert a plane wave carrying arbitrary polarization into spiral focusing beams with efficiency as high as 76% and 32% in transmission mode (*l* = 2) and in reflection mode (*l* = 4), respectively, and the relationship between the focus length in the two modes approximately satisfies ${ƒ}_{2}\approx 2.3{ƒ}_{1}$. In addition, the impact of Gaussian beams with different waist sizes on the MSF generator is demonstrated. The spiral focusing efficiency is proportional to the waist size of Gaussian beams and the SFB evolve into vortex beams when the waist size of Gaussian beams decreases. Owing to some advantages of the device we proposed, including wavelength thickness, high efficiency, tunability in topological charge and focal length, and insensitivity to polarization, our devices will be able to be applied to a variety of integrated compact optical systems, such as optical communication systems and optical trapping systems.

## Figures and Tables

**Figure 1 nanomaterials-10-01864-f001:**
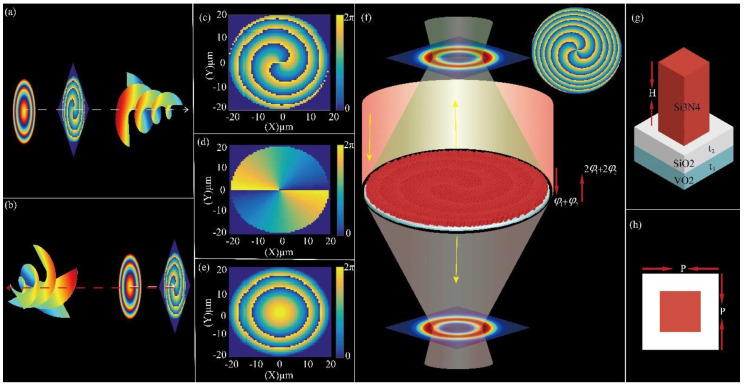
(**a**,**b**) Schematic diagram of metasurface spiral focusing (MSF) generator carrying topological charge of *l* = 2 and *l* = 4 work at transmission mode and at reflection mode, respectively. (**c**) Phase distribution of The MSF generator, which can be considered as the superimposition of the spiral phase distribution (**d**) and parabola phase distribution. (**e**,**f**) Schematic of a metasurface that generates spiral focusing beams (SFB) in both transmission mode and in reflection mode. Incident light with wavelength of 1550 nm illuminates on the metasurface along the negative *z*-axis and is focused into a focal ring. (**g**,**h**) Side view and top view of the unit cell structure. The optimized parameters are *P* = 652.5 nm, *H* = 1.65 µm, and $t_{1}=t_{2}=$ 500 nm.

**Figure 2 nanomaterials-10-01864-f002:**
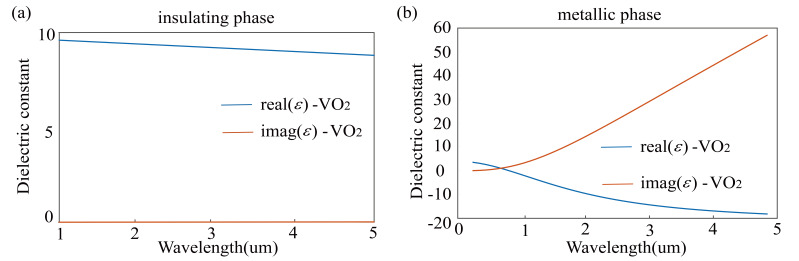
(**a**) Dielectric constant of VO_2_ in its low-temperature insulator phase. (**b**) Dielectric constant of VO_2_ in its high-temperature metallic phase.

**Figure 3 nanomaterials-10-01864-f003:**
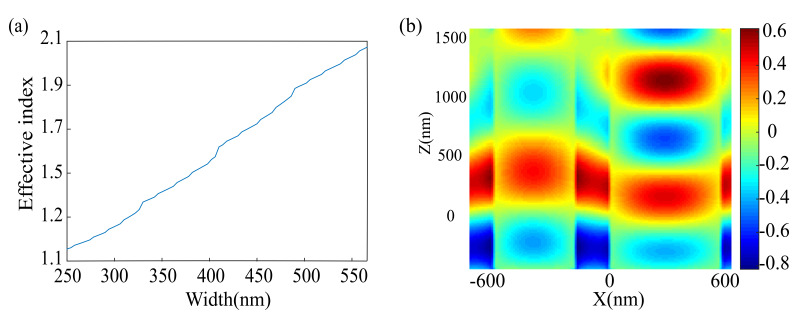
(**a**) Effective index of a single square silicon nitride (SSN) nanopillar as a function of its width (*w*) for x-LPL incidence ($E_{x}$). (**b**) Side views of the electric field (real [Ex]) distribution in two** SSN nanopillars with different widths (400 nm, 560 nm).

**Figure 4 nanomaterials-10-01864-f004:**
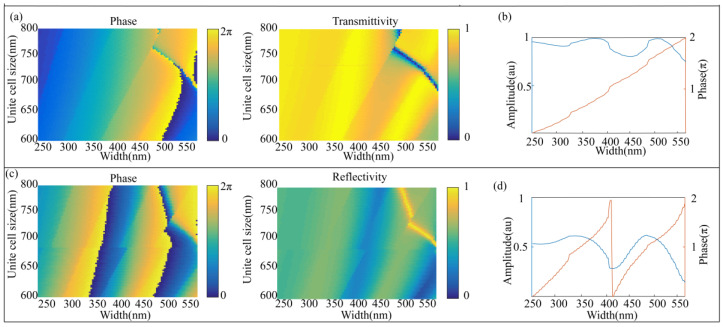
(**a**) Simulated transmittivity and phase of the transmitted field as a function of the unit cell size and the width of SSN nanopillar when VO_2_ is in the insulator phase. (**b**) Simulated transmittivity and phase of the transmitted field for a fixed unite cell size of 652.5 nm. (**c**) Simulated reflectivity and phase of the reflected field as a function of the unite cell size and the width of SSN nanopillar when VO_2_ is in the metallic phase. (**d**) Simulated reflectivity and phase of the reflected field for a fixed unite cell size of 652.5 nm. All the simulated results for (**a**–**d**) are performed at the wavelength 1550 nm with x-LPL incidence.

**Figure 5 nanomaterials-10-01864-f005:**
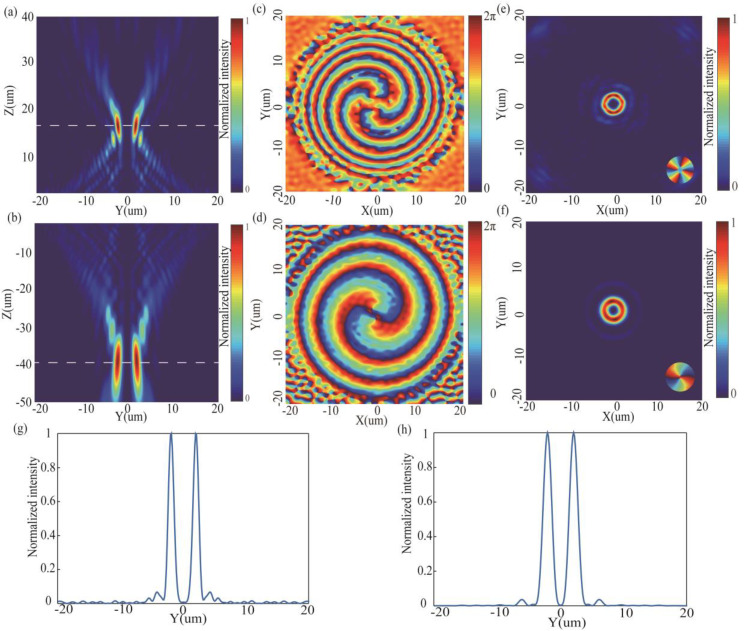
(**a**,**b**) Simulated intensity profiles on the y-z cross section for the MSF generator in reflection mode (*l* = 4) and in transmission mode (*l* = 2), respectively. (**c**,**d**) Simulated phase distribution of reflected light and transmitted light at one wavelength away from exit facet of the MSF generator, respectively. (**e**,**f**) Reflected focal ring and transmitted focal ring of the MSF generator at focal plane (z = 17.2 µm and z = −39 µm), respectively. (**g**,**h**) Corresponding vertical cuts of focal ring.

**Figure 6 nanomaterials-10-01864-f006:**
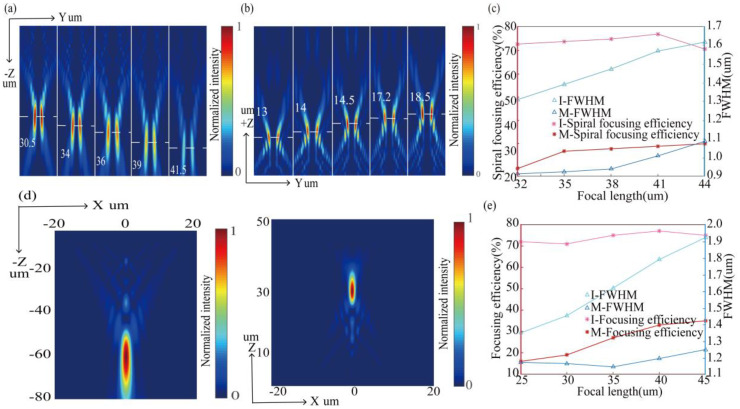
(**a**,**b**) Simulated intensity profiles on the y–z cross section of transmitted light and reflected light are produced by MSF generators with different focal lengths, respectively. (**c**) Simulated values of full width at half maximum (FWHM) and spiral focusing efficiency for several MSF generators with different focal lengths in two modes. (**d**) Simulated intensity profiles on thee y–z cross section for the MF generator in transmission mode and in reflection mode. (**e**) Simulated values of FWHM and focusing efficiency for several MF generators with different focal lengths in two modes.

**Figure 7 nanomaterials-10-01864-f007:**
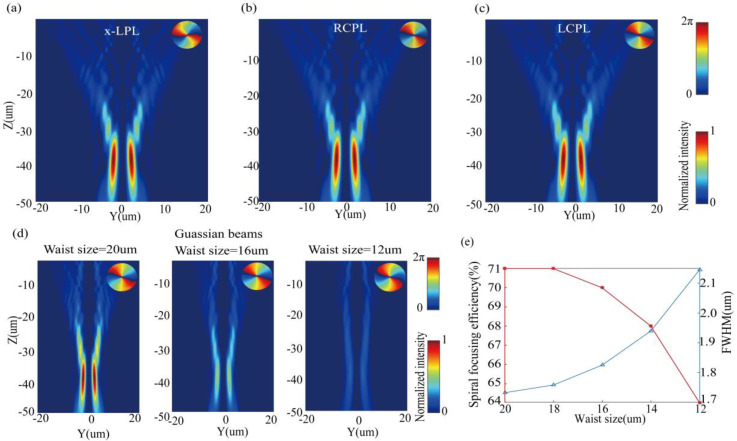
Simulated transmitted intensity profiles on the y–z cross section of the MSF generator for the normal incidences of (**a**) x-LPL, (**b**) right-handed circular polarization light, (**c**) left-handed circular polarization light. (**d**) Simulated transmitted intensity profiles on the y–z cross section of the MSF generator for the normal incidences of Gaussian beams with different waist sizes. (**e**) Simulated values of FWHM and spiral focusing efficiency for the normal incidences of Gaussian beams with different waist sizes.

**Figure 8 nanomaterials-10-01864-f008:**
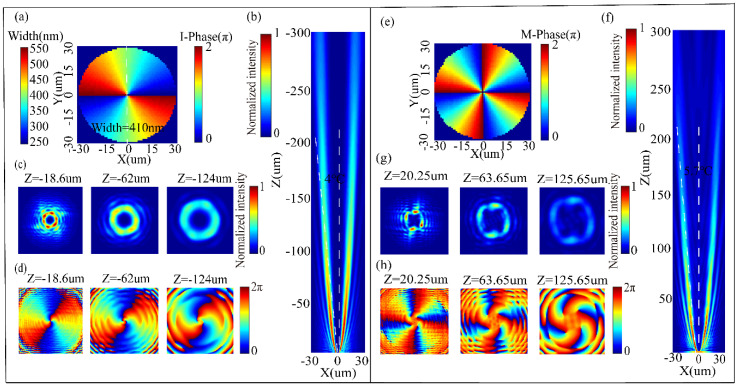
(**a**) Distribution of the calculated phase and width of nanopillar at the transmitted facet. (**b**) Simulated transmitted intensity profiles on the x–z cross section. (**c**) Simulated transmitted intensity profiles on the x–y cross section for $12\lambda_{i}$, $40\lambda_{i}$ and $80\lambda_{i}$ away from the exit facet of the metasurface. (**d**) Simulated transmitted phase distribution on the x–y cross section for $12\lambda_{i}$, $40\lambda_{i}$ and $80\lambda_{i}$ away from exit facet of the metasurface. (**e**) Calculated values of ideal phase distribution at the reflected facet. (**f**) Simulated reflected intensity profiles on the x–z cross section. (**g**) Simulated reflected intensity profiles on the x–y cross section for $12\lambda_{i}$, $40\lambda_{i}$ and $80\lambda_{i}$ away from exit facet of the metasurface. (**h**) Simulated reflected phase profiles on the x–y cross section for $12\lambda_{i}$, $40\lambda_{i}$ and $80\lambda_{i}$ away from the exit facet of the metasurface.

**Figure 9 nanomaterials-10-01864-f009:**
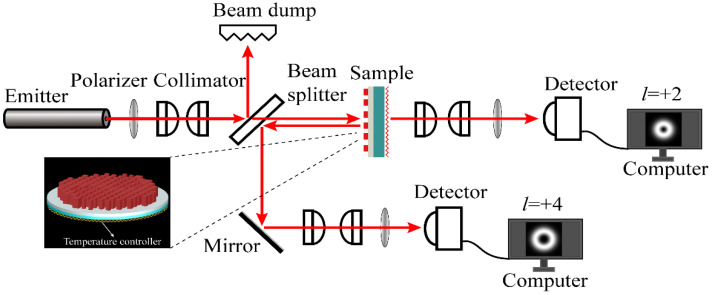
Schematic of the experimental setup.

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
