# Peer review of "Metasurface Spiral Focusing Generators with Tunable Orbital Angular Momentum Based on Slab Silicon Nitride Waveguide and Vanadium Dioxide (VO2)"

_nanomaterials, 2020, doi:10.3390/nano10091864_

Round 1

Reviewer 1 Report

The authors needs provide in the manuscript one section about the experimental setup. To understand the results is necessary this section.

the authors refers the use in optical communications  but not explain the aplications. Is possible using optical fibers?

the authors refers differents optical sources but not clarify  the differences and advantages.

Effect os temperature in the results.

I suggest a major review.

Author Response

Response to Reviewer 1 Comments

Point 1: The authors needs provide in the manuscript one section about the experimental setup. To understand the results is necessary this section.

Response 1: We are very grateful for your suggestions to modify the article. We have made appropriate changes based on your suggestions. We provided in the manuscript one section about the experimental setup (as described in “experimental feasibility”).

Point 2: The authors refers the use in optical communications, but not explain the applications. Is possible using optical fibers?

Response 2: In previous work, it has been reported to couple the optical vortex generated by the metasurface into the optical fiber [1]. The difference between our metasurface and [1] is that the phase modulation method is different, and the material is different, but the dimensions of the two metasurfaces are basically the same. The optical vortex generated by MSF generator can be coupled into optical fibers. Therefore, our devices will be able to applied to optical fibers communication (as the last sentence of the introduction).

Point 3: The authors refers differents optical sources but not clarify the differences and advantages.

Response 3: In this article, we have designed MSF generator that can work normally under linearly polarized light and circularly polarized light, Gaussian beams, which can greatly increase the practicability of the MSF generator. Generally speaking, the circular polarization light is generated by a superposition of two linearly polarized lights, whose polarization directions are perpendicular and which have phase differences of π/2. Gaussian beams are different from plane waves (linearly polarized light) because their light intensity gradually decreases along the radial direction. It is very important to study the influence of different light sources on metasurfaces. If a metasurface can work normally under any light source, it will inevitably reduce the complexity of the experimental measurement device. For example, because geometric phase-based metasurfaces usually only work under circularly polarized light, this means that the experimental measurement device based on the metasurfaces of geometric phase has one more quarter wave plate than the experimental measurement device based on the metasurfaces of propagation phase (such as the red text in “impact of the different light sources on MSF generator”).

Point 4: effect on temperature in the results.

Response 4: We also considered the influence of temperature on the MSF generator when designing the device. The transition of VO2 from the insulator state to the metallic state is a very fast process. Moreover, in the process of phase transition, VO2 is neither insulator state nor metallic state, and the real part and imaginary part of its dielectric constant are relatively close, so it does not have much research value. Generally speaking, Only the insulator state and metallic state of VO2 are considered [2-4]. In our research, we only consider the insulator and metallic states of VO2. Figure S1 shows the optical permittivity of VO2 [5], we can see that the temperature will significantly affect the optical permittivity of VO2 when the temperature is near the phase transition point (in the process of phase transition) and the temperature hardly affect the optical permittivity of VO2 when the temperature is far away from the phase transition point (in insulator state or metallic state). This means that when the temperature is far away from the phase transition point, small changes in temperature will not affect the final result. (such as the red text in the second paragraph of “theory and design”).

Figure S1. The (a) real, and (b) imaginary components of the optical permittivity of VO2

Thank you very much for reading our article in your busy time. No matter what the final result is, we wish you all the best. We tried our best to improve the manuscript and made some changes in the manuscript.

We appreciate for your warm work earnestly, and hope that the correction will meet with approval.

Once again, thank you very much for your comments and suggestions!

  1. Xi, W.; Jinwei, Z.; Jingbo, S.; Nezhad, V.F.; Cartwright, A.N.; Litchinitser, N.M. Metasurface-on-fiber enabled orbital angular momentum modes in conventional optical fibers. In Proceedings of 2014 Conference on Lasers and Electro-Optics (CLEO) - Laser Science to Photonic Applications, 8-13 June 2014; pp. 1-2.
  2. Chen, W.; Chen, R.; Zhou, Y.; Chen, R.; Ma, Y. Spin-dependent switchable metasurfaces using phase change materials. Opt. Express 2019, 27, 25678-25687.
  3. He, H.; Shang, X.; Xu, L.; Zhao, J.; Cai, W.; Wang, J.; Zhao, C.; Wang, L. Thermally switchable bifunctional plasmonic metasurface for perfect absorption and polarization conversion based on VO2. Opt. Express 2020, 28, 4563-4570.
  4. Mao, M.; Liang, Y.; Liang, R.; Zhao, L.; Xu, N.; Guo, J.; Wang, F.; Meng, H.; Liu, H.; Wei, Z. Dynamically Temperature-Voltage Controlled Multifunctional Device Based on VO2 and Graphene Hybrid Metamaterials: Perfect Absorber and Highly Efficient Polarization Converter. Nanomaterials 2019, 9, 1101.
  5. Butakov, N.; Valmianski, I.; Lewi, T.; Urban, C.; Ren, Z.; Mikhailovsky, A.; Wilson, S.; Schuller, I.; Schuller, J. Switchable Plasmonic-Dielectric Resonators with Metal-Insulator Transitions. ACS Photonics 2018, 5, 371-377.

Reviewer 2 Report

In the paper "Metasurface spiral focusing generators with tunable orbital angular momentum based on vanadium dioxide (VO2)" L. Chen et al deal with numerical simulation of metasurface, which converts linearly polarized incident light to optical vortex and focus it. The metalens works in two modes: reflection mode and transmission mode. The topological charge of the generated optical vortex was equal to 4 and 2, respectivelly.  

The paper is interesting; however, there are some mistakes that should be corrected before publication of the manuscript:

1) The title is frustrating, because the investigated metalens is not based on vanadium dioxed. It is based on SiN nanopilars, and VO2 layer was added for reflection mode realization.

2) The manuscript does not contain details of numerical simulation: software, parameters, etc.

3) Figure 1f shows that focal spot in reflection mode is located on the optical axis; however Figure 1b shows that it is shifted from the axis

4) Similar metalenses, whitch converts light to the focused optical vortex, were investigated in [APL. v. 114, p. 141107 (2019); J. Opt. v. 21, p. 055004 (2019)]

5) What is the diameter of the lens?

Author Response

Point 1: The title is frustrating, because the investigated metalens is not based on vanadium dioxide. It is based on SSN nanopillars, and VO2 layer was added for reflection mode realization.

Response 1: Thank you very much for your suggestion, the title has been changed to “Metasurface spiral focusing generators with tunable orbital angular momentum based on slab silicon nitride waveguide and vanadium dioxide (VO2)”.

Point 2: The manuscript does not contain details of numerical simulation: software, parameters, etc.

Response 2: We are very sorry for our negligence. We have added the details of numerical simulation in the manuscript (such as the red text in the third paragraph of “Theory and design”). The numerical simulation of the unit cell structure is performed using the commercially available three-dimensional finite difference time domain (FDTD) solver from Lumerical Inc. The size of a mesh cell is 0.01 µm.

Point 3: Figure 1f shows that focal spot in reflection mode is located on the optical axis, however, Figure 1b shows that it is shifted from the axis.

Response 3: We are very sorry for our negligence. We have appropriately modified the problems in Figure 1b.

Point 4: Similar metalenses, whitch converts light to the focused optical vortex, were investigated in [APL. v. 114, p. 141107 (2019); J. Opt. v. 21, p. 055004 (2019)].

Response 4: Subwavelength grating-based spiral metalens for tight focusing of laser light. Applied Physics Letters 2019, 114, 141107. In this article, they investigate a 16-sector spiral metalens consisting of subwavelength gratings. This metalens converts linearly polarized light into an azimuthally polarized optical vortex and focuses it. Their metalens is based on subwavelength and only can generate one OAM mode. Single metalens for generating polarization and phase singularities leading to a reverse flow of energy. Journal of Optics 2019, 21, 055004. In this article, they investigate metasurface-based optical element composed of a set of subwavelength diffraction gratings, whose anisotropic transmittance is described by a matrix of polarization rotation by angle m, where is the polar angle, generate an mth order azimuthally or radially polarized beam, when illuminated by linearly polarized light, or an optical vortex with topological charge m, when illuminated by circularly polarized light. Their metalens is based on optical element composed of a set of subwavelength diffraction gratings and only can generate one OAM mode for one polarized light. The local phase modulation is achieved by rotating the polar angle of the subwavelength diffraction gratings. Therefore, the optical vortex can only be produced when the incident light is circularly polarized light. However, in our article, we design a highly efficient MSF generator based on the phase mechanism of slab waveguide theory and vanadium dioxide. This MSF generator can generate spiral focusing beams with different topological charges both in transmission mode and reflection mode by controlling the phase change of vanadium dioxide. Furthermore, the MSF generator insensitivity to polarization.

Point 5: What is the diameter of the lens?

Response 5: We are very sorry for our negligence. The diameter of the lens is 20um (such as the red text in “the relationship between the focal lengths in two modes”).

Thank you very much for reading our article in your busy time. No matter what the final result is, we wish you all the best. We tried our best to improve the manuscript and made some changes in the manuscript.

We appreciate for your warm work earnestly, and hope that the correction will meet with approval.

Once again, thank you very much for your comments and suggestions!

Round 2

Reviewer 1 Report

The authors made changes in the manuscript in response to previous reports. I recommend accept in present form.

Author Response

Dear Reviewer

Thanks very much for your kind work and consideration on publication of our paper. On behalf of my co-authors, we would like to express our great appreciation to editor and reviewers.

Thank you and best regards.

Reviewer 2 Report

The manuscript could be published; however, some minor comments could be taken into account:

1. In the author's response I read that "...The diameter of the lens is 20um "; however, in the manuscript the authors write that the radius of the lens is equal to 20 μm. Which value is correct?

2. Instead of "My device can overcome ..." it would be better to use "Our device"

Author Response

Point 1:  In the author's response I read that "...The diameter of the lens is 20um "; however, in the manuscript the authors write that the radius of the lens is equal to 20 μm. Which value is correct?

Response 1: We are very sorry for our negligence. The diameter of the lens is 40um

Point 2: Instead of "My device can overcome ..." it would be better to use "Our device"

Response 2: We are very grateful for your suggestions to the article. We made appropriate changes based on your suggestions.

This manuscript is a resubmission of an earlier submission. The following is a list of the peer review reports and author responses from that submission.

Round 1

Reviewer 1 Report

The paper by Chen et al. deals with numerical study of a metamaterial acting as a vortex beam generator based on metamaterials in VO2.

This subject itself is topical but it is approached in a very unclear way. First, figures are very small and have a poor quality. Some notations are inadequate (for example “transmission” instead of “transmittivity” or “transmission coefficient” as in Fig.2d). In addition, phase values are restricted, in all figures, to be between 0 and 2pi or –pi to pi which reveals non-physical discontinuities (see function “unwrap” in Matlab environment). Other important details exist and make reading and understanding the article very difficult.

The first oddity I noted lies in the calculation of the reflectivity of a VO2 layer in metallic phase. For this, a Drude model seems to be used to model the dispersion properties. However, the most shocking thing is to see that the authors did their calculation with a numerical method (the FDTD for instance) which is completely devoid of any physical sense because such a calculation is analytical and can be done easily.

In general, we cannot understand what the authors have done as numerical simulations because they almost never give all the parameters of their modeled structures. For example, taking the paragraph starting at line 108, an effective index calculation is carried out as a function of the width of the Si3N4 nanopillar. It is not indicated whether the modeled structure consists of a single square section waveguide of Si3N4 or a periodic structure of such waveguides. Which guided mode is considered? For which polarization? From figure 2c we can guess that calculations are done for periodic structure!!!! But it still is an assumption. Is the coupling between nearest nanopillars is taken into account? In addition, the discontinuities that appear on the transmission spectra of Fig.2d, f and h are never discussed. Do they correspond to a known physical phenomena?

For all these reasons, I cannot recommend the publication of this paper in its present form. A great effort must be done to make it accessible to the reader.

Reviewer 2 Report

The paper considers generation of vortices of different orders using metasurfaces. Authors have suggested using phase transition of vanadium dioxide for these purposes and have shown transformation efficiency of more than 72%. According to the results of numerical simulation, any polarization state can be transformed into vortices using the suggested design.

The results are of interest for applications in optical communication technology in order to reach more dense information capacities.

Reviewer 3 Report

The authors present one topic interesting: orbital angular momentum. However, the authors present only theoretical work. In conclusions the authors say "In this article, a highly efficient MSF generator with insensitivity to polarization is proposed, which can realize switch metallic and dielectric metasurfaces. The MSF generator are able to convert  a plane wave carrying arbitrary polarization into spiral focusing beams with efficiency as high as 76%
270 and 32% in transmission mode (l=2) and in reflection mode (l=4),...".

This statement is not entirely correct. the authors  presents only the theoretical part. What is the reason for not presenting an experiemental part?   The legend of figures are extend. Some parts can be included in the main text.   Some equation need references.   The introduction need improve.   The authors not explain clearly the effect of temperature in the final results   Some experimental setupt and results using this study need to be included.   In the present version, i recommend reject. The authors needs provide a new version.